# High-Dose Intermittent Treatment with the Multikinase Inhibitor Sunitinib Leads to High Intra-Tumor Drug Exposure in Patients with Advanced Solid Tumors

**DOI:** 10.3390/cancers14246061

**Published:** 2022-12-09

**Authors:** Sophie L. Gerritse, Mariette Labots, Rob Ter Heine, Henk Dekker, Dennis Poel, Daniele V. F. Tauriello, Iris D. Nagtegaal, Erik Van Den Hombergh, Nielka Van Erp, Henk M. W. Verheul

**Affiliations:** 1Department of Medical Oncology, Radboud Institute for Health Sciences, Radboud University Medical Center, Geert Grooteplein Zuid 10, 6525 GA Nijmegen, The Netherlands; 2Department of Medical Oncology, Cancer Institute, Erasmus University Medical Center, Doctor Molewaterplein 40, 3015 GD Rotterdam, The Netherlands; 3Department of Medical Oncology, Cancer Center Amsterdam, Location VUmc, De Boelelaan 1118, 1081 HV Amsterdam, The Netherlands; 4Department of Pharmacy, Radboud Institute for Health Sciences, Radboud University Medical Center, Geert Grooteplein Zuid 10, 6525 GA Nijmegen, The Netherlands; 5Department of Cell Biology, Radboud Institute for Molecular Life Sciences, Radboud University Medical Center, Geert Grooteplein Zuid 10, 6525 GA Nijmegen, The Netherlands; 6Department of Pathology, Radboud University Medical Center, Geert Grooteplein Zuid 10, 6525 GA Nijmegen, The Netherlands

**Keywords:** high-dose, intermittent, sunitinib, tumor concentrations

## Abstract

**Simple Summary:**

Multitargeted tyrosine kinase inhibitors (TKIs) provide clinical benefit in patients with cancer when daily dosed to continuously inhibit their designated targets. Alternative dosing strategies pursuing higher tumor drug concentrations may improve their benefit by inhibiting relevant off-target kinase activity. We studied the tumor-drug exposure in a previously established safe and feasible high-dose intermittent treatment strategy of the multikinase inhibitor sunitinib in relation to its treatment benefit.

**Abstract:**

Patients with advanced cancer refractory to standard treatment were treated with sunitinib at a dose of 300 mg once every week (Q1W) or 700 mg once every two weeks (Q2W). Tumor, skin and plasma concentrations were measured and immunohistochemical staining for tumor cell proliferation (TCP), microvessel density (MVD) and T-cell infiltration was performed on tumor biopsies before and after 17 days of treatment. Oral administration of 300 mg sunitinib Q1W or 700 mg Q2W resulted in 19-fold (range 5–35×) and 37-fold higher (range 10–88×) tumor drug concentrations compared to parallel maximum plasma drug concentrations, respectively. Patients with higher tumor sunitinib concentrations had favorable progression-free and overall survival than those with lower concentrations (*p* = 0.046 and 0.024, respectively). In addition, immunohistochemistry of tumor biopsies revealed an induction of T-cell infiltration upon treatment. These findings provide pharmacological and biological insights in the clinical benefit from high-dose intermittent sunitinib treatment. It emphasizes the potential benefit from reaching higher tumor drug concentrations and the value of measuring TKI tumor- over plasma-concentrations. The finding that reaching higher tumor drug concentrations provides most clinical benefit in patients with treatment refractory malignancies indicates that the inhibitory potency of sunitinib may be enforced by a high-dose intermittent treatment schedule. These results provide proof of concept for testing other clinically available multitargeted tyrosine kinase inhibitors in a high-dose intermittent treatment schedule.

## 1. Introduction

Protein kinases promote cell proliferation, survival and migration of cancer cells [1]. Over 50 kinase inhibitors, including multitargeted tyrosine kinase inhibitors (TKIs), have been approved [1,2,3]. One of them is sunitinib, an inhibitor of the vascular endothelial growth factors VEGFR1 (FLT1), VEGFR2 (FLK1/KDR), fetal liver tyrosine kinase receptor 3 (FLT3), stem-cell growth factor receptor (KIT), and the platelet-derived growth factor receptors PDGFRα and PDGFRβ [4]. It is approved for oral dosing once daily (Q1D) at its maximum tolerated dose (MTD) of 50 mg (4 weeks on/2 weeks off) [5].

Despite significant clinical benefit in advanced renal cell cancer, imatinib-resistant or -intolerant gastrointestinal stromal tumors and well-differentiated neuroendocrine tumors of the pancreas, treatment benefit is of limited duration due to acquired resistance. Moreover, for most common tumor types, such as colorectal cancer, prostate cancer, and breast cancer, sunitinib treatment is ineffective in the majority of patients [2].

Sunitinib is extensively distributed into various tissues, whereas only a small fraction of the drug remains in the circulation, reflected by a relatively large apparent volume of distribution (2230 L) [5]. It is metabolized by cytochrome P450 3A4 to the equipotent active metabolite N-desethyl sunitinib (SU12662) [4]. Combination of sunitinib + N-desethyl sunitinib represents the total active drug in plasma (SUM concentration), because N-desethyl sunitinib has a similar inhibitory profile and plasma protein binding to that of sunitinib in vitro [6]. Sunitinib and its metabolite are small hydrophobic molecules that can efficiently cross cellular membranes and enter tumor and tumor-supporting cells. 

The hypothesis behind its daily dosing strategy is that a steady sunitinib plasma concentration will enable continuous inhibition of its designated targets [4]. Proof for this concept is lacking and knowledge regarding its exact mechanism of action at the tumor and tumor-supporting cells is limited even as the drug resistance mechanisms to sunitinib. Our recent studies indicate that drug concentrations reached in the tumor are underestimated by sunitinib plasma concentrations. Gotink et al. reported 10-times higher tumor concentrations compared to plasma in mice and 30-times higher tumor sunitinib concentrations (9.5–10.9 micromolar (µM) compared to plasma concentrations at the standard daily schedule in three patients [7]. Labots et al. confirmed these findings in patients, reporting median sunitinib tumor concentrations of 9.0 µM compared to median plasma concentrations of 0.1 µM (3.586 versus 40 ug/L) [8]. Finally, Rovithi et al. measured high sunitinib tumor concentrations for both the standard daily schedule (13.5 μM (range 7.0–37.4 μM)) and a high-dose once every week (Q1W) schedule (36 μM (range: 7.8–130 μM)) in tumors growing on the chorioallantoic membrane of the chicken embryo (CAM-assay) [9]. Based on these findings, we hypothesized that the antitumor activity of sunitinib can be improved by increasing tumor-drug exposure to significantly inhibit additional relevant (off-target) kinases with a distinct—and potentially more favorable—composite efficacy [7,8]. We established a safe and feasible high-dose intermittent schedule of 300 mg sunitinib once every week (Q1W) and 700 mg sunitinib once every two weeks (Q2W) in patients with advanced cancer [10]. Here, we measured tumor, skin and plasma concentrations of this high-dose intermittent treatment strategy with sunitinib in patients with various advanced cancer types. We focused our analyses on potential relations of these concentrations with treatment benefit and biological activity. 

## 2. Materials & Methods

Patients and study design: From 2013–2018, a dose escalation phase I, single-institution clinical trial, was conducted at Amsterdam University Medical Center, location VUmc, Amsterdam, the Netherlands (ClinicalTrials.gov identifier 02058901) to establish an intermittent dosing schedule of sunitinib (Q1W or Q2W). All patients provided written informed consent before entering the study. The study was conducted in accordance with the Declaration of Helsinki and with the principles of the International Conference on Harmonization Guidelines for Good Clinical Practice. The study protocol was approved by the local institutional board and an independent Ethics Committee.

Sunitinib was administered orally according to one of two schedules: Q1W or Q2W. A standard Phase 1 “3 + 3 design” was used with a starting dose of 200 mg sunitinib administered orally once every week and escalating in steps of 100 mg. Patients continued treatment until progression, intolerance, or withdrawal of consent. The primary outcomes of the phase I clinical trial were described previously [10]. 

Food: An extra cohort was opened to evaluate the effect of food on the interpatient variation in bioavailability and to study whether concomitant intake with food could further increase the maximum plasma concentration (C_max_)_._ (See Appendix A for details regarding the food cohort)

Study pharmacokinetic assessment in plasma: Ethylenediaminetetraacetic acid (EDTA) blood was collected pretreatment and subsequently at multiple time points in all patients (0, 2, 4, 6, 8, 10, and 24 h post-dose on day 1 for both time schedules and thereafter at days 3, 8, 10, 15, 17, and 22 for the once weekly schedule and at days 3, 15, 17, and 29 for the once every 2 weeks schedule). All samples were centrifuged within 1 h after collection at 2000 g for 10 min (at 4 degrees Celsius (°C)), plasma was separated and stored at −80 °C until the day of analysis. 

Study pharmacokinetic assessment in tumor and skin tissue: Patients who gave additional informed consent for tissue collection, underwent a tumor needle biopsy before and after 17 days of study treatment taken by an interventional radiologist. An optional skin biopsy (from healthy skin tissue at the abdomen) was taken after 17 days of treatment. All tissue samples were immediately snap frozen (within 1 min) in liquid nitrogen and cryopreserved at −80 °C until the day of analysis. 

Tissue biopsies were weighed accurately before analysis. Tissue homogenate was prepared by using a mechanical homogenizer. The tumor tissue was homogenized with the precellys lysing kit (bertin instruments^®^) in the Roche MagNA lyzer^®^. Proteins in the homogenate were precipitated with methanol primed with internal standards (the stable isotopes 2H10-sunitinib and 2H5-desethylsunitinib). Subsequently, sunitinib and N-desethyl sunitinib were quantified against a calibration curve with known concentrations. Tissue homogenate samples were directly analyzed. 

Drug concentration measurements: Sunitinib and N-desethylsunitinib were measured in all samples; using a validated liquid chromatography–tandem mass spectrometry (LC-MS-MS) as previously described (see Appendix A). 

Immunohistochemistry: Immunohistochemical staining was performed on 3–5 μm thick sections of the (freeze) tumor biopsies for CD3, CD31 and Ki67 according to standard methods (see Appendix A). Morphological characteristics were assessed in hematoxylin-eosin (H&E)-stained tissue sections. Evaluation of staining was performed by four investigators using light microscopy of serial tissue sections. Ki-67 was scored based on the percentage of positively stained malignant nuclei, using the following ranges: 0% to 20%, >20% to 40%, >40% to 60%, >60% to 80%, and >80% to 100%. Semi-quantification of CD31 staining was scored manually by a pathologist. The number of cells positive for CD3, CD31 was expressed as an estimated proportion of the total number of cells.

Statistics: Descriptive statistics were used to summarize patient characteristics and sunitinib and N-desethyl sunitinib concentrations. Two-sided paired Student’s *t*-test on log-transformed data was used to compare concentrations between sunitinib and N-desethyl sunitinib concentrations in plasma, tissue, and skin tissue and pre- and on treatment immunohistochemistry analysis. An independent *t*-test on log-transformed data was used to compare sunitinib concentrations between the two MTD-dose levels (300 mg Q1W and 700 mg Q2W). Sunitinib and N-desethyl sunitinib concentrations in plasma, tumor and skin and their correlations were calculated using the Spearman’s rank correlation coefficient (ρ). Efficacy was assessed in patients who completed at least two weeks of treatment. Kaplan–Meier curves and log-rank tests were used to evaluate differences in PFS and OS between two groups dichotomized by geometric mean sunitinib and N-desethyl sunitinib concentration. Sunitinib and N-desethyl sunitinib concentrations in plasma, tumor and skin and their correlation with PFS and OS were calculated using the Spearman’s rank correlation coefficient (ρ). Statistical significance was set at *p* < 0.05. All statistical analyses were performed in SPSS, version 25.

Pharmacokinetic analysis: The concentration–time data for sunitinib and N-desethyl sunitinib were analyzed by means of nonlinear mixed-effects modeling (NONMEM, V7.4.3). Since there is no evidence for the exact relationship between plasma and tumor concentrations, we chose to use the concentration–time data calculated by NONMEM for C_max_, the average concentration over time (C_average_) and the trough concentrations just before the next dose was administered (C_trough_) based on rich PK-data of each individual patient. As a starting point for the analysis, the previously developed population pharmacokinetic model for sunitinib and its active metabolite by Yu et al. was used [6]. The model was extended with a sequential zero- and first-order absorption model, to describe the dissolution and subsequent absorption of the high sunitinib dose in the gastrointestinal tract. Inter-occasion variability in bioavailability and absorption duration were allowed in the model. The control stream of the final model and the goodness-of-fit plots are supplied in the Appendix A. The developed model was used to derive the individual predictions for the C_max_, C_average_, and C_trough_. SUM concentrations were calculated.

## 3. Results

Baseline Patient Characteristics and Covariates: Out of 83 patients with refractory solid tumors participating in the phase 1 trial, 21 and 29 patients were treated with sunitinib at the established maximal tolerated dose (MTD) of Q1W 300 mg and Q2W 700 mg, respectively. Characteristics of all participating patients, including a flowchart are provided in Figure 1. Results for all individual patients are shown in Appendix A. All participating patients have progressed and died at time of analysis.

Tumor drug concentration: To determine tumor drug concentrations with this treatment schedule, non-mandatory tumor biopsies from 22 patients at day 17 after start of treatment were obtained (after 3 cycles Q1W or 2 cycles Q2W). Twenty of 22 tumor biopsies were taken from patients treated at the MTD of 300 mg Q1W or 700 mg Q2W. The geometric mean of the sunitinib + N-desethyl sunitinib (SUM) tumor concentration of patients treated with 300 mg Q1W (n = 12) was 6.656 µg/L (coefficient of variation (CV) 111%) versus 17.376 µg/L (CV 109%) in those treated with sunitinib 700 mg Q2W (n = 8, *p*-Value (*p*) = 0.07) (Table 1 and Appendix A). The other two patients received a dose of 400 mg Q1W and 800 mg Q2W resulting in tumor drug concentrations of 1.369 and 515 µg/L respectively. 

Tumor concentrations obtained from patients treated with 300 mg sunitinib Q1W were approximately 19-times higher than achieved maximum plasma concentration (C_max_) (geometric mean of 6656 µg/L, range 877–48.647 µg/L versus 349 µg/L, range 183–1396 µg/L, respectively; *p* = 0.001) and approximately 3 times higher than skin concentrations (a geometric mean of 1957 µg/L, range 319–8772 µg/L; *p* = 0.001) on day 17. Tumor concentrations obtained from patients treated with 700 mg sunitinib Q2W were approximately 37-times higher than achieved plasma C_max_ (geometric mean of 17.376 µg/L, range 2791–69.781 µg/L versus 473 µg/L, range 283–790 µg/L, respectively; *p* = 0.001) and approximately 9-times higher than skin concentrations (a geometric mean of 1.857 µg/L, range 379–19.539 µg/L; *p* = 0.001) on day 17 (Table 1). No correlation was observed between tumor concentrations and plasma- or skin concentrations and food did not affect the oral uptake of sunitinib (Figure 2A and Appendix A).

Tumor drug concentrations in relation to outcome: As previously reported, median progression free survival (PFS) for all patients was 2.5 months (range 0.5–11 months); 2.2 months for 300 mg Q1W (range 0.7–11) and 2.6 months (range 0.5–11) for 700 mg Q2W. Median overall survival (OS) for all patients was 4.9 months (range 0.8–27 months); 3.1 months for 300 mg Q1W (range 1.2–15) and 4.9 months for 700 mg Q2W (range 1.3−19.6) [10]. The SUM concentration of sunitinib + N-desethyl sunitinib in tumor biopsies correlated positively with PFS (Spearman’s rank correlation coefficient (ρ) = 0.43, *p*-Value 0.046) and OS (ρ 0.49; *p*-Value 0.024) (Figure 2B). No correlation between PFS and OS with either plasma or skin concentrations were found (Appendix A).

Immunohistochemical staining: Immunohistochemical staining for tumor cell proliferation (TCP), microvessel density (MVD) and T-cell infiltration was performed in 10 available paired tumor biopsies before (at baseline) and after 17 days of treatment (Figure 2A). While no significant differences in pre- versus on treatment biopsies were found for TCP and MVD, a significant increase in T cell infiltration, assessed by CD3 staining, was found (paired *t*-test, mean increase 3.55 (from 8.6% to 12.2%), 95% CI 1.15–5.95, *p*-Value 0.009) (Figure 3 and Appendix A).

## 4. Discussion

We herein report remarkably high tumor drug exposure from high-dose intermittent sunitinib treatment in patients with advanced cancer lacking standard treatment options. A more favorable PFS and OS were observed in patients with the highest sunitinib + N-desethyl sunitinib tumor concentrations. These tumor sunitinib concentrations are 2–5-times higher than reported for the tumor concentrations derived from daily treatment [8]. Importantly, these concentrations exceeded the concentration required for inhibition of its ‘designated angiogenesis-related targets’ indicative for potential inhibition of a wide relevant spectrum of the human kinome (Appendix A) [4,5,11,12,13]. Clinical trials with other TKIs (e.g., erlotinib and sorafenib) at high-dose intermittent treatment schedules also showed higher plasma C_max_ concentrations compared to daily dosing, but no data on their tumor concentrations have been reported [14,15,16].

The significantly higher tumor drug concentrations compared to plasma drug concentrations might be due to the chemical characteristics of sunitinib, trafficking easily through membranes in its neutral state but trapped in its protonated form under acidic circumstances (in cellular vesicles) [7]. We measured total sunitinib tumor tissue concentrations, while only the intracellular unbound, available free drug can interact and inhibit multiple kinases [11]. Determination of the exact fraction of intracellular unbound, available free drug from the total drug concentration is not feasible since no assays are available to measure intracellular unbound drug concentrations. When considering this relative shortcoming, it is striking that a positive relation between the total tumor drug concentration and clinical outcome was found. This supports our hypothesis that high dosing contributes to inhibition of additional relevant ‘off’ targets in tumor and supporting cells of the tumor microenvironment [12,17,18]. Sunitinib has been implicated in impairing T cell activation and proliferation in vitro and in vivo. Guislain et al. found improved tumor-infiltrating lymphocyte expansion in tumor digests of renal cell cancer patients who were pre-treated with sunitinib. This was associated with reduced intratumoral myeloid-derived suppressor cells. Exploratory analyses of 10 paired tumor biopsies in our population showed a significant increase in T-cell infiltration over time. This increase provides a potential lead for combination strategies with immune checkpoint inhibitors and high-dose intermittent sunitinib [18,19].

In conclusion, this study reveals that high-dose intermittent treatment with sunitinib leads to high tumor drug concentrations compared to prior reported tumor drug concentrations of daily dosing. These high tumor drug concentrations are favorably related to an improved PFS and OS in a small and heterogeneous group of patients with advanced cancer. This improved outcome might be due to additional off-target kinase inhibition with a distinct—and potentially more favorable—composite efficacy. The results of this study emphasize the importance of measuring drug concentrations at the target site and underscores the limitations of regular plasma pharmacokinetics in determining optimal doses and schedules. Further evaluation of a potential benefit from high-dose intermittent sunitinib treatment compared to standard dosing is of interest in registered indications such as renal cell cancer and gastrointestinal stroma cell tumors (GIST), and in potential new indications such as the current ongoing trial in colorectal cancer (NCT03909724).

## Figures and Tables

**Figure 1 cancers-14-06061-f001:**
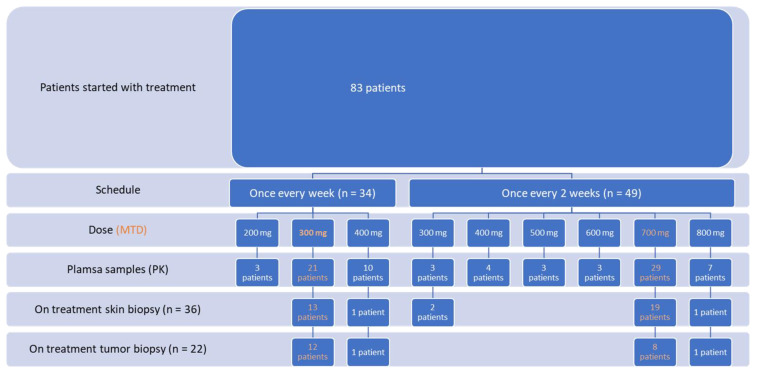
Flowchart.

**Figure 2 cancers-14-06061-f002:**
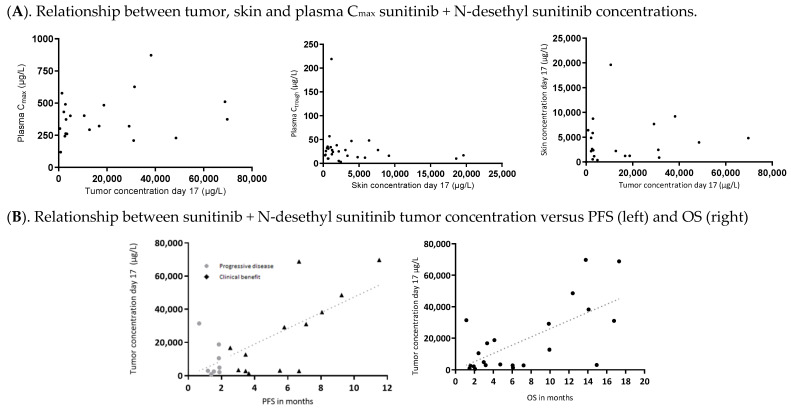
(**A**), Relationship between C_max_ plasma sunitinib + N-desethyl sunitinib and corresponding tumor concentration and C_max_ and corresponding skin concentration. Relationship between C_max_ plasma sunitinib + N-desethyl sunitinib and corresponding tumor concentration (**left**); C_max_ plasma sunitinib + N-desethyl sunitinib and corresponding skin concentration (**middle**) and sunitinib + N-desethyl sunitinib skin concentration at day 17 and corresponding tumor drug concentration at day 17 (**right**). All concentrations represent the sunitinib + N-desethyl sunitinib sum concentration of an individual patient. The triangles represent patients with clinical benefit (stable disease of response) at first CT-scan evaluation. (**B**), Correlation between sunitinib + N-desethyl sunitinib tumor concentration versus PFS (**left**) and OS (**right**). All concentrations represent the sunitinib + N-desethyl sunitinib sum concentration of an individual patient. The SUM concentration of sunitinib + N-desethyl sunitinib in tumor biopsies significantly correlated with PFS (Spearman’s rank correlation coefficient (ρ) = 0.43, *p*-Value 0.046) and OS (ρ 0.49; *p*-Value 0.024). All participating patients have progressed and died at time of analysis. Abbreviations: PK, Pharmacokinetic; µg/L, microgram per Liter.

**Figure 3 cancers-14-06061-f003:**
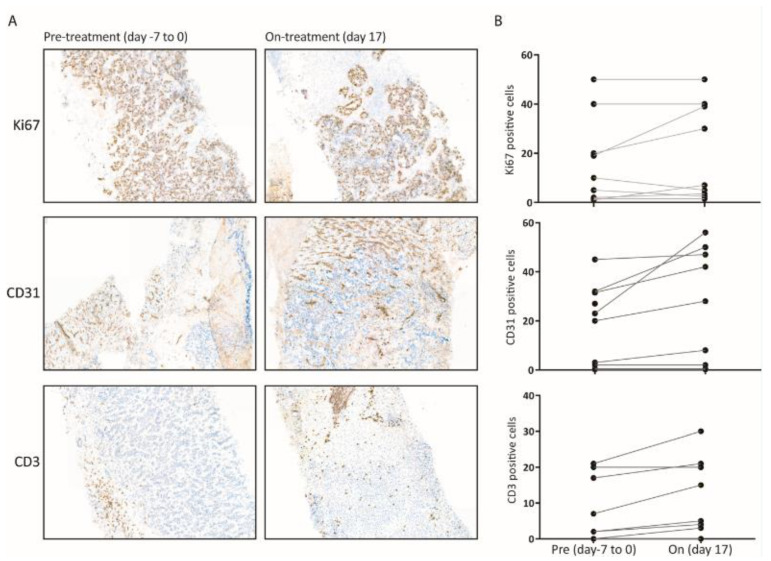
(**A**) Shows ×10 magnification of representative immunohistochemical staining (pre-treatment (day 7–0) and on-treatment (day 17)) of a tumor treated as indicated. Upper panel: Ki−67 staining (cholangiocarcinoma). Medium panel: CD31 staining (colorectal cancer), lower panel: CD3 staining (breast cancer). (**B**) Quantification of tumor cell proliferation using CD31, Ki67 and CD3 staining on pre- and on (day 17) treatment biopsies. Data are expressed as percentage of Ki-67 positive tumor cells to total tumor cells per field (×40 magnification). CD31 and CD3 are expressed as the number of positive cells per field (×40 magnification for CD3 and ×10 magnification for CD31). Similar colors represent the same patient.

**Table 1 cancers-14-06061-t001:** Mean Pharmacokinetic Parameters of Intermittent High-Dose Sunitinib at the MTD dose levels (300 mg Q1W and 700 mg Q2W).

	Tumor (n = 20)		Skin (n = 32)		Plasma (n = 50)						Plasma (Literature) (n = 9) [5]
	Concentration (µg/L) day 17				C_max_ (µg/L)		C_trough_ (µg/L)		C_average_ day 0–17 (µg/L)		C_max_ (µg/L)
	300 mg Q1W (n = 12)	700 mg Q2W (n = 8)	300 mg Q1W (n = 13)	700 mg Q2W (n = 19)	300 mg Q1W (n = 21)	700 mg Q2W (n = 29)	300 mg Q1W (n = 21)	700 mg Q2W (n = 29)	300 mg Q1W (n = 21)	700 mg Q2W (n = 29)	50 mg once a day
Geometric mean	6656	17376	1957	1857	349	473	43	19	118	158	108
CV%	111	109	86	75	70	28	85	63	57	31	
Mean	14238	28825	3016	4034	407	491	54	23	133	166	
Range	877–48,647	2791–69,781	319–8772	379–19,539	183–1396	283–790	16–219	4–76	80–275	60–359	

Abbreviations: C_average_, average concentration; C_max_, peak concentration; C_trough_, trough concentration; CV, coefficient of variation; MTD, maximum tolerated dose; SD, standard deviation; SUM, sunitinib + N-desethyl sunitinib sum concentration; µg/L, microgram per Liter.

## Data Availability

The authors declare that all data used in the conduct of the analyses are available within the article and (Appendix A) tables and figures. To protect the privacy and confidentiality of patients in this study, clinical data are not publicly available in a repository or in the Appendix A of the article, but they can made available upon reasonable request to the corresponding author. Those requests will be reviewed by a study steering committee to verify whether the request is subject to any intellectual property or confidentiality obligations. All data shared will be de-identified.

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
