# Peer review of "High-Dose Intermittent Treatment with the Multikinase Inhibitor Sunitinib Leads to High Intra-Tumor Drug Exposure in Patients with Advanced Solid Tumors"

_cancers, 2022, doi:10.3390/cancers14246061_

Round 1

Reviewer 1 Report

My main comment is about the  the method to assess the relationship between intratumoral sunitinib concentration and efficacy.

To illustrate the relationship between intratumoral sunitinib concentration you present the rgression curve with the spearman correlation coefficient between various sunitinib concentration (intratumoral -skin , plasma) and PFS and OS

First question : usually PFS and OS data contain censored data thus you can't perform a correlation test. How did you manage censored data or maybe all patients have progressed and died ? This point should be clarified.

Second : you used Cmax to illustrate the absence of correlation between plasma exposure and survival. But since a relationship between outcomes and Cmin or AUC but not Cmax  have been published for sunitinib, these othersexposure determination should be tested instead of Cmax. 

In the method section you say that the population has been dichotomized in two group according to the geometric mean sunitinib concentration, then the difference in PFS and OS hase been tested between these two group by the log-rang test. I did not see the kaplan meier curves nor the results of the log-rank tests.

Other comments :

About patients’ characteristics, in table 1 you mention 83 patients treated whereas in the published phase 1  Rovithi et al show a 73 patients population from whom 71 have been treated with intermittent sunitinib.

You talk about an heterogenous population, the pathologies deserve to be presented

In fig 1, as in fig supplemental data5 the regression curves deserve to be drawn for all graphs, to better visualize the trends in correlation.

You present the relationship between sunitinib concentration and efficacy but nothing about toxicity. Since you reach very high exposure levels, this point should be commented : does very high level of sutinib concentration were associated with an increasing risk in toxicity?

Reviewer 2 Report

The manuscript was aimed to show the benefits of high-dose intermittent therapy of the multi-kinase inhibitor sunitinib for patients with diverse solid tumors (colorectal, breast, lung cancer, etc.).

The major concerns about the manuscript are the following:

1) the patients enrolled in this study represent heterogenous population (regarding the name of disease). Moreover the size of the primary tumor is not shown in the Supplementary Table 1 illustrating the characteristics of  the patients. All together,  these might have significant impact on the intra-tumor accumulation of sunitinib.

2) The manuscript has the duplications. Data shown in Figure 1A is duplicated in Supplementary Data 1. Similarly, data shown in Figure 1B was also included into the Supplementary Data 5.  

3) The number of cases of PFS and OS shown in Figure 1B is different. Similar, the number of events shown in the Figure 1A (left, right and in the middle) are different from each other. 

4) No impact of sunitinib on the tumor cell proliferation and microvessel density observed by IHC-staining might be also due to the diversity of the tumors included in the present study. Therefore, I advice to look over these IHC-based parameters in each particular group of patients and/OR increase the number of patients with each particular tumor (e.g. CRC, breast, lung cancer, etc.). 

 5) Despite the heterogenicity of the patients enrolled in this study, the authors observed the induction of T-cell infiltration upon sunitinib treatment. Despite this fact is intriguing, discussion lacks any information about this finding.  I suggest the  authors to explain the potential immunomodulatory (off-target) effects of sunitinib (similar to the well-known effect of imatinib mesylate) and include the corresponding references. 

Reviewer 3 Report

The authors provide the clinical significance of sunitinib treatment for solid tumor patient and, pharmacological and biological data which is very necessary to the public health.

This manuscript is for the most part well written with substantial discussion of results and postulated according to the evidence provided. The review organization is impressive, and the figures provided were comprehensive. The references are appropriate and timely.

Minor criticisms

• Please undergo a thorough check of the manuscript for typographical and grammatical errors.

Round 2

Reviewer 2 Report

The authors responded to the comments and suggestions. 

The quality of the manuscript was improved and it be accepted for publication in present form.